# Attitudes to the use of animals in biomedical research: Effects of stigma and selected research project summaries

**Helen J. Cassaday** [1] *, **Lucy Cavenagh**[1], **Hiruni Aluthgamage**[1], **Aoife Crooks**[1], **Charlotte Bonardi**[1], **Carl W. Stevenson**[2], **Lauren Waite**[1], **Charlotte Muir**[1,3]

**1** School of Psychology, University of Nottingham, Nottingham, United Kingdom, **2** School of Biosciences, University of Nottingham, Nottingham, United Kingdom, **3** School of Physiology, Pharmacology, and Neuroscience, University of Bristol, Bristol, United Kingdom

* helen.cassaday@nottingham.ac.uk

**Data Availability Statement:** All relevant data are within the paper and its Supporting Information files, and are also deposited in the University of

## Abstract

Three groups of participants (largely recruited from the UK) completed a survey to examine attitudes to the use of animals in biomedical research, after reading the lay (N = 182) or technical (N = 201) summary of a research project, or no summary (N = 215). They then completed a survey comprising the animal attitude (AAS), animal purpose (APQ), belief in animal mind (BAM) and empathy quotient (EQ) scales. The APQ was adapted to assess attitudes towards the use of animals for research into disorders selected to be perceived as controllable and so 'blameworthy' and potentially stigmatised (addiction and obesity) and 'psychological' (schizophrenia and addiction) versus 'physical' (cardiovascular disease and obesity), across selected species (rats, mice, fish pigs and monkeys). Thus, the APQ was used to examine how the effects of perceived controllability and the nature of the disorder affected attitudes to animal use, in different species and in the three summary groups. As expected, attitudes to animal use as measured by the AAS and the APQ (total) correlated positively with BAM and EQ scores, consistent with the assumption that the scales all measured pro-welfare attitudes. Participants in the two research summary groups did not differentiate the use of rats, mice and fish (or fish and pigs in the technical summary group), whereas all species were differentiated in the no summary group. Participants given the lay summary were as concerned about the use of animals for schizophrenia as for addiction research. APQ ratings otherwise indicated more concern for animals used for addiction research (and for obesity compared to cardiovascular disease in all summary groups). Therefore, the information provided by a research project summary influenced attitudes to use of animals in biomedical research. However, there was no overall increase in agreement with animal use in either of the summary groups.

## Introduction

Attitudes to the use of animals in biomedical research are influenced by a host of cultural factors, from economic metrics such as a country's gross domestic product [1], to more individual

Nottingham Research Data Repository (doi: 10.
17639/nott.7305), to be made freely available post-publication.

**Funding:** This work was in part supported by the Biotechnology and Biological Sciences Research Council https://www.ukri.org/councils/bbsrc/ [grant number BB/S000119/1] awarded to HJC, CWS and CB. The funders had no role in study design, data collection and analysis, decision to publish, or preparation of the manuscript.

**Competing interests:** This study examines the effectiveness of project summaries outlining a plan of work supported by the Biotechnology and Biological Sciences Research Council [grant number BB/S000119/1] as moderators of attitudes to animal use. The BBSRC had no further role in the study. The authors have declared that no individual competing interests exist.

factors such as rural versus urban background and personality [2]. Secondary analyses of data from national surveys conducted across Europe showed that (amongst other factors) the level of scientific information provided on new medical discoveries was positively associated with the acceptance of animal experimentation [1]. However, systematic evidence on the influence of information, such as that provided by positive media coverage and public engagement activities, is lacking [1, 2]. Barriers to the general public's lay understanding of animal research include the accessibility of communications provided by biomedical scientists, which may relate to the lack of training, as well as the perception of risk from animal activism [3].

Lay summaries of science should in principle provide a cost-effective resource for promoting public understanding [4–6]. Summaries of publicly funded research are widely available and openly accessible online, for example, via UK Research and Innovation https://gtr.ukri.org/, but there is little evidence as to their effectiveness in promoting the public understanding of science in general, or in mitigating prejudices pertaining to areas of biomedical research. To address this gap, the present study examined the effects of reading selected summaries on attitudes towards animal use in the context of lay beliefs about controllability, blameworthiness and the stigma that may be associated with different health conditions.

## Stigma and blame

Stigma involves the exclusion or discrimination of someone based on a label attributed to them, for example an illness [7–9]. Stigma may prevent those affected from contacting the appropriate services for help, as well as reduce the public contribution towards research funding for mental health [10]. Stereotyped negative preconceptions of those with mental illness are found even amongst mental health professionals [8], and medical and psychology students [11]. Psychoeducational interventions (information brochures and video presentations) have decreased aspects of negative stereotyping [11], and more positive attitudes towards mental than physical illness have since been reported in studies of mental health practitioners [12]. Still, those who experience medical conditions which may be misconceived as having an element of personal responsibility, such as addiction and obesity, are discriminated against by society [13].

In addition to the stigmatisation of mental health problems in general, there are particular biases towards specific disorders. There is a level of individual control associated with blameworthy disorders that individuals are perceived to 'bring on themselves', on the assumption that individuals can control their environmental exposures. Drug addicts tend to be considered responsible for their actions rather than mentally ill, even though research has suggested heritable risk factors [14, 15]. Obesity too has heritable risk factors [16, 17]. However, similar to addiction, the physical disorder of obesity is perceived as caused by the individual, thus having negative connotations of blame resulting in stigmatized attitudes [18].

In the present study, the effects of stigma and blame were examined by comparing attitudes to animal use for medical research into addiction and obesity versus schizophrenia and cardio-vascular disease (CVD). We also examined the consistency of these attitudes across different animal species.

## Animal species

All laboratory work with animals is highly regulated, for example, by the UK Home Office *Animals* (Scientific Procedures) Act (1986) and in the EU (since 1986) by Directive 2010/63/EU (superseding Directive 86/609/EEC) on the protection of animals used for scientific purposes. Yet two thirds of the UK public do not feel well-informed about animal research practices, and some strongly held beliefs are incorrect, for example, that cosmetics testing is still permitted in

the EU and UK. With regard to the type of animal and the nature of the research, rats and mice are viewed as the most acceptable species for medical research purposes [19], and are indeed the most commonly used species in the research regulated and monitored by the UK Home Office [20]. In studies of addiction, schizophrenia and obesity, the use of rodents is widespread. However, non-mammalian species such as fish are increasingly used, for example to study CVD [21]. Pigs have been used to study obesity and CVD because of the similarities to humans in their digestive tract and cardiovascular system [22, 23]. Non-human primates may still be used in biomedical research, due to their close evolutionary proximity to humans, but with correspondingly greater public concern about such usage [19, 24, 25]. In the present study, we examined attitudes towards the use of rats, mice, fish, pigs and monkeys, all of which may be used in medical research, using the animal purpose questionnaire (APQ) [24, 26].

Effects of species have been demonstrated in a number of previous studies with the APQ [24, 26, 27]. For example, participants show relatively less disagreement with the use of rats, mice and fish [26], and more disagreement with the use of monkeys [24, 26]. Agreement with the use of pigs has depended on the purpose of use in studies examining a range of purposes wider than medical research, for example, for food production [26].

We also examined belief in animal mind (BAM) for the same selected species [28, 29] and other potential determinants of (dis)agreement with animal use: Herzog's animal attitude scale (AAS) [30, 31]; and the empathy quotient (EQ) [32].

## Aims and hypotheses

The primary aim was to systematically test the effects of an intervention commonly assumed to influence the public understanding of science, comparing the effects on attitudes to animal use of reading the lay summary of a research project with those of reading the technical summary of the same research project. In addition, we compared attitudes towards animal use for research on biomedical disorders which differ in public perceptions of controllability (to the extent they may be viewed as 'self-inflicted' and blameworthy), as well as in relation to distinctions drawn between 'physical' versus 'psychological' disorders.

The predictions to be tested were that: (1) exposure to a research summary (particularly the lay version intended to be more accessible) will reduce disagreement with animal use for biomedical research; (2) there will be relatively more support for the use of animals for research on conditions viewed as physical rather than psychological; and (3) levels of agreement with the use of animals will be lower for research into conditions that are perceived as controllable (addiction and obesity) than for those that are not (schizophrenia and CVD).To test for differences by species, we examined self-reported attitudes for the purpose of biomedical research on each of the four disorders, for each of five species used in such research.

The present study used a factorial design, enabling us to identify how the effects of exposure to a research summary (lay or technical) depended on attributes of the disorders examined (within the context of medical research), as well as on the species in use for this research. The factorial design enabled us to test for differential effects by species; for example, only rats were specifically mentioned in the research summaries. Similarly, the summaries might be expected to have a differential effect on attitudes towards the use of animals for psychological rather than physical disorders, because the research described was psychological (though the control of food intake and obesity were mentioned in the technical and lay summaries, respectively). We also examined whether the predicted effect of perceived controllability in reducing agreement with animal use for more 'blameworthy' disorders would be mitigated by exposure to the lay summary, which specifically mentioned the implications of the research for our understanding of addiction and obesity (as well as for schizophrenia).

## Methods

### Ethics

Ethics approval was obtained from the University of Nottingham UK School of Psychology Ethics Committee (Ref: 994R and Ref: S1021). Participants were first presented with the invitation to participate, information on the study and contact details in the event of complaint (the Chair of the University of Nottingham UK School of Psychology Ethics Committee). Consent was confirmed immediately below by written checkbox Yes/No confirmations in response to the following series of questions: (1) 'Are you aged 18 or over?'; (2) 'Have you read and understood the above information?'; (3) 'Do you feel like you have the opportunity to ask questions about the study if you choose?'; (4) 'Do you understand that you are free to withdraw from the study? (at any time and without giving a reason)'; (5) 'I give permission for my data from this study to be shared with other researchers provided that my anonymity is completely protected'; and (6) 'Do you agree to take part in the study?' Participants who checked 'No' in response to any of the above questions could not progress to complete the survey.

Authors had no access to information that could identify individual participants during or after data collection, just a nickname was provided in case any participant wished to request removal of their data. Participants were also given the specific instruction: 'Please provide a memorable unique identifier, that is **NOT** related to your name. Please remember this identifier as it will aid in identifying the appropriate data, should you wish to withdraw yours from the study.'

### Participants

Participants were recruited through word of mouth, using friends, family, social media networks and the website 'Survey Circle'. The links were shared via broadly similar but non-overlapping social networks (to reduce the possibility of responses to more than one survey). The surveys for the different summary groups were not available at the same time on Survey Circle. Data collection was completed November 2021 to March 2022.

Of the total of 598 participants who started the study, 178 participants identified as male and 383 participants as female. In total 467 participants identified as omnivore and 76 participants as non-meat eating (vegetarian or vegan). In total 204 participants reported a relevant degree and 369 participants no relevant degree. The subsequent statistical analyses included participants who completed the survey at least to the end of the APQ questions. For the no summary group, the total number of participants was 253 (reduced to 215 after incomplete APQ entries were removed), of average age 29.71 (SEM = 0.917), in the range 18–75 years. For the lay summary group, 221 participants were recruited (reduced to 182 completing the APQ), of average age 30.36 (SEM = 0.978), in the range 18–69 years. For the technical summary group, the 320 participants recruited (201 after removing incomplete entries) were of average age 29.22 (SEM = 0.965), in the range 18–67 years. The other demographic data are summarised by summary type group in Table 1.

### Design

The APQ scores were analysed using mixed design analysis of variance (ANOVA) to test the predictions. The between-subjects factor was summary with 3 levels (none, lay, technical). There were 3 within-subjects factors: species at 5 levels (rats, mice, fish, pigs and monkeys); the perceived control or 'blameworthy' nature of the disorder at 2 levels (controllable, uncontrollable); and the nature of the disorder, again at 2 levels (psychological, physical). The mixed factorial design is complex (3 x 5 x 2 x 2), but predictions pertain to main effects and hypotheses

**Table 1. Participant demographics as percentages by survey summary type.**

| Survey summary type | | | | |
|---|---|---|---|---|
| **Demographic** | | **None (%)** | **Lay (%)** | **Technical (%)** |
| Eating orientation | Omnivore | 61.8 | 69.4 | 68.8 |
| | Pescatarian | 4.8 | 3.3 | 4.8 |
| | Flexitarian | 13.5 | 10.0 | 7.5 |
| | Vegetarian | 12.1 | 10.0 | 10.8 |
| | Vegan | 2.4 | 3.9 | 1.1 |
| | Other/Prefer not to say | 5.3 | 3.3 | 7.0 |
| Gender | Male | 21.7 | 28.3 | 44.1 |
| | Female | 77.8 | 67.8 | 53.8 |
| | Other/Prefer not to say | 0.5 | 3.9 | 2.2 |
| Relevant degree | Yes | 43.5 | 38.9 | 23.7 |
| | No | 56.5 | 61.1 | 76.3 |

*Note*: percentage (%) reported characteristics for valid responses (of participants who in addition completed at least the APQ) by summary type: None (no summary; valid N = 205); Lay (lay summary; valid N = 180) and Technical (technical summary; valid N = 185).

about the effects of species, tested by examination of 2-way and 3-way interactions in this model. The 4-way interaction was of no *a priori* interest. Three further ANOVAs (2 x 3 x 5 x 2 x 2) were conducted to test for interactions between summary and confounded demographic factors (gender, eating orientation and educational experience). Details of further statistical analyses are provided below.

## Materials

The surveys were administered using the Qualtrics platform. Each survey comprised four questionnaires (presented in the order listed below) designed to measure participants' attitudes to animal use and/or pro-welfare characteristics, as well as a set of four demographic questions, regarding age, gender, eating orientation and any (current or previously undertaken) psychology, neuroscience, biological or medical sciences degree. The questionnaires were scored such that in each case high scores indicated pro-welfare attitudes to animal use.

The surveys administered to the 3 groups were identical except for the research project summary (absent, lay or technical; S1 Appendix), which was presented on a separate screen before the questionnaires, and seven additional questions introduced at the end of the two summary surveys (presented after the matched presentation of the standard questionnaires and demographic questions). This 'your views' section was added to get direct ratings of agreement with (question 1) the justification of animal research for 'psychological' as compared to physical illnesses or conditions, and (question 2) the justification of animal research for illnesses or conditions that people seem to 'bring on themselves'. Participants were also asked to provide perceived control ratings for each of the disorders examined (questions 3–6), as well as ratings to indicate their direct experience of these disorders in friends or family members (question 7).

## Summaries

The summaries were published and openly available as the lay versus technical research summaries of a UK-funded research project. The study presented here was intended to assess the effectiveness of these summaries in moderating attitudes to animal use without further editing (except the removal of some text from the lay summary, in order to match the length of the

technical summary, and to present as a single paragraph as per the technical summary). The summaries and their source are provided in the S1 Appendix. The three summary types (none, lay and technical) formed the independent variable.

## Animal attitude scale (AAS)

The AAS was the first dependent variable measure completed, using the short 10-item adaptation [31]. The AAS asks participants to rate their agreement with general statements concerning human use of animals. A 5-point Likert scale was used to score participants' answers, with 'strongly disagree' coded 1 and 'strongly agree' coded 5, with 3 as the neutral point of 'undecided'. Half of the questions (2, 3, 4, 7 and 8) were reverse scored. Possible scores were in the range 1–5 per item, with total scores in the range 10–50 for the 10-item scale. The AAS showed good internal consistency across the 10 items ($\alpha = 0.774$).

## Animal purpose questionnaire (APQ)

The APQ was modified to examine attitudes to animal use for 4 specific research purposes across 5 selected species. The selected purposes were 'addiction research', 'schizophrenia research', 'obesity research' and 'cardiovascular disease research.' The five selected species were rats, mice, fish, pigs and monkeys, presented in a randomised order for comparative ratings by purpose (with purposes listed in a fixed order, to reduce the risk of confusion in the compressed presentation format used; Fig 1). Participants rated their degree of agreement on a 5-item Likert scale, ranging from 'strongly disagree' (coded 5) to 'strongly agree' (coded 1), with 3 being the neutral point, for the five species in turn, each across the four different research purposes, forming in total 20 questions to be completed by the participants. A higher total score showed a lower agreement with use of animals in research, consistent with higher levels of concern for animal wellbeing. Possible scores were in the range 1–5 per item, with total scores in the range 20–100 for the 20-item scale. The APQ showed excellent internal consistency across the 20 items ($\alpha = 0.980$). These APQ ratings provided the second set of dependent variables.

## Belief in animal mind (BAM)

The original BAM questionnaire [28, 29] was adapted to present the original four questions with each of the five species included in the APQ (rats, mice, fish, pigs and monkeys), giving 20 items in total. A 7-point Likert scale was used, ranging from 'strongly agree' (coded 1) to 'strongly disagree' (coded 7), with 3 being the neutral point, for participants to rate their level of agreement with each statement. Both the order of the questions and the order of species within each question were randomised and presented in a compressed format [24]. Two questions (2 and 3) were reverse scored to avoid response bias. A higher score relates to a greater belief in animal mind, typically associated with higher levels of concern for animal wellbeing. Possible scores were in the range 1–7 per item, with total scores in the range 20–140 for the 20-item scale. The BAM showed excellent internal consistency across the 20 items ($\alpha = 0.906$). The BAM total provided the third dependent variable.

## Empathy quotient (EQ)

The original 40-item EQ was used [32]. The measure consists of 40 key items and 20 filler questions. We followed the scoring instructions provided by the original authors of this questionnaire [32]. Participants rated their responses on a 4-point Likert scale from 'strongly agree' to 'strongly disagree'. For half the key items (1, 6, 19, 22, 25, 26, 35, 36, 37, 38, 41, 42, 43, 44, 52,

**To what extent do you agree with the use of RATS for the following purposes:**

|  | Strongly Agree | Agree | Neutral | Disagree | Strongly Disagree |
|---|---|---|---|---|---|
| Addiction research | ○ | ○ | ○ | ○ | ○ |
| Schizophrenia research | ○ | ○ | ○ | ○ | ○ |
| Obesity research | ○ | ○ | ○ | ○ | ○ |
| Cardiovascular disease research | ○ | ○ | ○ | ○ | ○ |

**To what extent do you agree with the use of PIGS for the following purposes:**

|  | Strongly Agree | Agree | Neutral | Disagree | Strongly Disagree |
|---|---|---|---|---|---|
| Addiction research | ○ | ○ | ○ | ○ | ○ |
| Schizophrenia research | ○ | ○ | ○ | ○ | ○ |
| Obesity research | ○ | ○ | ○ | ○ | ○ |
| Cardiovascular disease research | ○ | ○ | ○ | ○ | ○ |

**To what extent do you agree with the use of MICE for the following purposes:**

|  | Strongly Agree | Agree | Neutral | Disagree | Strongly Disagree |
|---|---|---|---|---|---|
| Addiction research | ○ | ○ | ○ | ○ | ○ |
| Schizophrenia research | ○ | ○ | ○ | ○ | ○ |
| Obesity research | ○ | ○ | ○ | ○ | ○ |
| Cardiovascular disease research | ○ | ○ | ○ | ○ | ○ |

**To what extent do you agree with the use of FISH for the following purposes:**

|  | Strongly Agree | Agree | Neutral | Disagree | Strongly Disagree |
|---|---|---|---|---|---|
| Addiction research | ○ | ○ | ○ | ○ | ○ |
| Schizophrenia research | ○ | ○ | ○ | ○ | ○ |
| Obesity research | ○ | ○ | ○ | ○ | ○ |
| Cardiovascular disease research | ○ | ○ | ○ | ○ | ○ |

**To what extent do you agree with the use of MONKEYS for the following purposes:**

|  | Strongly Agree | Agree | Neutral | Disagree | Strongly Disagree |
|---|---|---|---|---|---|
| Addiction research | ○ | ○ | ○ | ○ | ○ |
| Schizophrenia research | ○ | ○ | ○ | ○ | ○ |
| Obesity research | ○ | ○ | ○ | ○ | ○ |
| Cardiovascular disease research | ○ | ○ | ○ | ○ | ○ |

**Fig 1. The presentation format of the animal purpose questionnaire (APQ) used to examine attitudes towards the use of rats, pigs, mice, fish and monkeys, for research into addiction, schizophrenia, obesity and cardiovascular disease, in three groups of participants.** The research purposes were always in the same fixed order for each question. The order of presentation of the questions by species was randomised.

54, 55, 57, 58, 59, 60) those who responded 'strongly agree' scored 2 points and 'slightly agree' scored 1 point. The remaining key items (4, 8, 10, 11, 12, 14, 15,18, 21, 27, 28, 29, 32, 34, 39, 46, 48, 49, 50) were reverse scored: those who answered 'strongly disagree' scored 2 points, and those who answered 'slightly disagree' scored 1 point. A total score between 0 and 32 reflects

lower than average levels of empathy; scores between 33 and 52 indicate average levels of empathy; scores between 53 and 63 suggest above average empathetic ability; and scores higher than 64 are taken to reflect very high levels of empathy. Total scores were in the range 0–80. The test-retest reliability for the EQ has previously been reported to be very good ($r = 0.97$) [32]. The EPQ total provided the fourth dependent variable.

## Demographic information

Participants were asked to provide the following demographic information: gender identity (male, female, other, prefer not to say); age in years; eating orientation (omnivore, pescatarian, flexitarian, vegetarian, vegan, other, prefer not to say); and whether they had achieved or were undertaking a degree in psychology, neuroscience or another biological or medical sciences subject (yes or no).

## Individual perceptions

In the final 'your views' section, participants were asked to rate on a 5-point Likert scale (from strongly agree to strongly disagree) their level of agreement with the following statements: 'Research with animals is less justified for disorders that people seem to bring on themselves'; 'The health risks of addiction/ schizophrenia/ obesity/ cardiovascular disease are beyond the individual's control'; and 'People close to me have been affected by the disorders mentioned in this survey (addiction, schizophrenia, obesity and/or cardiovascular disease)'. Strongly agree was coded 1 and strongly disagree was coded 5, with 3 being the neutral point. Thus, high scores reflected views that research was justified, that disorders were controllable and/or reflected no personal experience of these disorders.

## Procedures

Before completing the questionnaires, participants were presented with the information screen. The instructions included a brief overview of what to expect (a series of questionnaires about attitudes to animals and animal use, plus some requested background information), followed by the checkbox confirmations of their understanding of the study and consent to participate. Participants were also advised that to complete the study would take 15–25 minutes (no summary), or 20–30 minutes (for each of the survey versions preceded by a summary).

The final screen provided a debrief, describing the aims and hypotheses of the research. Participants were provided with the researchers' e-mail addresses in the event they had any questions about the study or wished their data to be removed (based on the nickname provided and date of completion). Contact details for independent help and advice from the Samaritans were also provided at the bottom of the screen, in case participants felt that any form of distress had been triggered by the survey.

## Data analyses

The data were analysed using IBM SPSS Statistics 28.0, by ANOVA for the APQ, as described above. Obesity and addiction were coded controllable, and schizophrenia and CVD as uncontrollable. Obesity and CVD were coded as physical disorders, and addiction and schizophrenia as psychological disorders.

Multivariate analyses were used to examine participant age and questionnaire totals, to see if these were matched across the summary type groups. Chi-square tests of independence were used to compare other (categorical) participant demographics across the summary groups. As some of the demographic sub-groups were very small, analyses by gender were restricted to

male versus female. Eating orientation was coded as meat-eating (including pescatarian and flexitarian) versus non-meat eating (vegan and vegetarian). Educational experience was coded as the presence or absence of relevant degree-level university study.

The simple main effects pairwise comparisons used to explore interactions were Bonferroni corrected as appropriate, and the adjusted $p$ values are reported in the text. Age was a continuous variable used in correlational analyses. Pearson's correlations were also conducted to investigate relationships between total APQ scores (across species and disorder), total AAS, BAM and EQ scores (and age).

The data are provided in the S2 Appendix and in the University of Nottingham Research Data Repository—doi: 10.17639/nott.7305.

## Results

Mauchly's test of sphericity indicated that the assumption of sphericity was violated for species, species by psychological (vs physical) nature of disorder, species by controllability of disorder (and the three-way interaction between these factors, NS, not reported). Greenhouse-Geisser corrected values (degrees of freedom to the nearest integer) are reported as applicable for the factorial analyses [33].

Shapiro-Wilk tests of normality were failed, $p < 0.05$, for AAS, APQ, BAM and the EQ scores across the 3 summary groups, with two exceptions, for BAM total no summary, $p = 0.051$, and EQ total technical summary, $p = 0.096$. However, non-normality is less of a problem in large samples and the Type 1 error rate is relatively unaffected by non-normality [34, 35].

There was homogeneity of variances between groups as assessed by Levene's test for equality of variance, for the AAS, BAM and the EQ scores, $p > 0.05$. Homogeneity of variances was violated for the APQ total scores, $p = 0.012$. However, the survey groups had roughly equal sample sizes, and the relatively large sample sizes mitigate the non-homogeneity [35]. Moreover, Pearson correlations are robust against violations of assumptions [36].

### Differences in participant characteristics across the summary type groups

The relation between gender and survey summary group allocation was significant, $\chi^2(2,561) = 24.615$, $p < 0.001$. As shown in Table 1, the surveys were completed by more females than males, but these proportions were more evenly matched in the case of the technical summary group (completed by 82 males and 100 females). The relation between relevant degree-level education and summary allocation was significant $\chi^2(2,573) = 18.027$, $p < 0.001$. The majority of participants did not have relevant degree-level education, but a smaller proportion of those reading the technical summary before completing the survey reported relevant degree-level education, compared to those in the no summary or lay summary groups. However, eating orientation was not different by summary group allocation, $\chi^2(2,543) = 0.359$, $p = 0.836$.

Multivariate analyses showed that participants completing the surveys were also very well matched for age, $F(2,567) = 0.342$, $p = 0.710$, $\eta_p^2 = 0.001$, and there was no effect of summary group on any of the total scores for the APQ, AAS, BAM or EQ, maximum $F(2,567) = 1.656$, $p = 0.192$, $\eta_p^2 = 0.006$, for the AAS. Thus, the overall levels of pro-welfare tendencies were similar across the summary groups, and there was no evidence that reading the summaries had any general effects on measures of pro-welfare attitudes.

Multivariate analyses of the 'your views' items showed that participants reading the technical summary before completing the survey reported higher agreement that the health risks of obesity, $F(1,272) = 6.150$, $p = 0.014$, $\eta_p^2 = 0.022$, and CVD, $F(1,272) = 3.997$, $p = 0.047$, $\eta_p^2 = 0.014$, are beyond the individual's control, compared to the levels of agreement indicated by

participants in the lay summary group. There were otherwise no differences between the technical and lay summary groups on the ratings given to these items.

## Relationships between APQ, AAS, BAM and EQ scores and between APQ and controllability ratings

As expected, given that they all measure pro-welfare tendencies, the totals for the APQ (across species and disorder), AAS, BAM and EQ scores were all positively correlated (Table 2). Age was not significantly correlated with any of the questionnaire scores, maximum $r(570) = 0.081$, $p = 0.054$.

The post-questionnaire agreement ratings that the health risks of each of the disorders are beyond the individual's control (reflecting the view that some disorders were less blameworthy) showed positive correlations with the corresponding APQ ratings averaged by disorder across species, for addiction $r(337) = 0.177$, $p = 0.001$, and schizophrenia, $r(319) = 0.198$, $p < 0.001$, but not for obesity, $r(335) = 0.020$, $p = 0.718$, or CVD, $r(328) = 0.100$, $p = 0.069$.

## Differences by species and research purpose of use (measured by the APQ) across summary type groups

As shown in Fig 2A, there was relatively greater disagreement with the use of pigs and monkeys. There was a main effect of species, $F(3,1614) = 130.505$, $p<0.001$, $\eta_p^2 = 0.180$. There was no main effect of summary type, $F(2,595) = 1.172$, $p = 0.310$, $\eta_p^2 = 0.004$, but there was an interaction between summary and species, $F(8,2380) = 3.882$, $p < 0.001$, $\eta_p^2 = 0.013$, so we consider species differences in relation to summary group. In the absence of any summary, participants' attitudes towards using the species examined were (without exception) all significantly different from each other. After reading the lay summary, there was higher disagreement with using pigs and monkeys than rats, mice and fish, all $p < 0.001$, but attitudes towards the use of rats, mice and fish were not different from each other. After reading the technical summary, participants showed higher disagreement with the use of monkeys than all other animals. However, attitudes towards the use of fish were now not different from attitudes towards the use of pigs. Attitudes towards the use of rats and mice, and mice and fish, were not different from each other. For full details of the pairwise comparisons, please see S3 Appendix Pairwise Comparisons (A).

Fig 2B shows how attitudes to animal use related to research purpose by species. Statistically, there was an interaction between species and controllability of disorder, $F(3,2075) = 2.501$, $p = 0.049$, $\eta_p^2 = 0.004$. However, the pairwise comparisons by controllability were individually significant for all species, $p < 0.001$.

**Table 2. The correlations between questionnaire measures of pro-welfare attitudes.**

| Scale | AAS | APQ | BAM | EQ |
|-------|-----|-----|-----|-----|
| AAS | - | < .001 | < .001 | < .001 |
| APQ | 0.602 | - | < .001 | < .001 |
| BAM | 0.340 | 0.306 | - | .004 |
| EQ | 0.168 | 0.141 | 0.121 | - |

*Note*: Pearson correlation coefficients (lower left) and uncorrected $p$ values (upper right) for total scores on the animal attitudes scale (AAS), animal purpose questionnaire (APQ), belief in animal mind (BAM) and empathy quotient (EQ) scales. Bonferroni-corrected critical $p$ value = 0.008. N = 579–598.

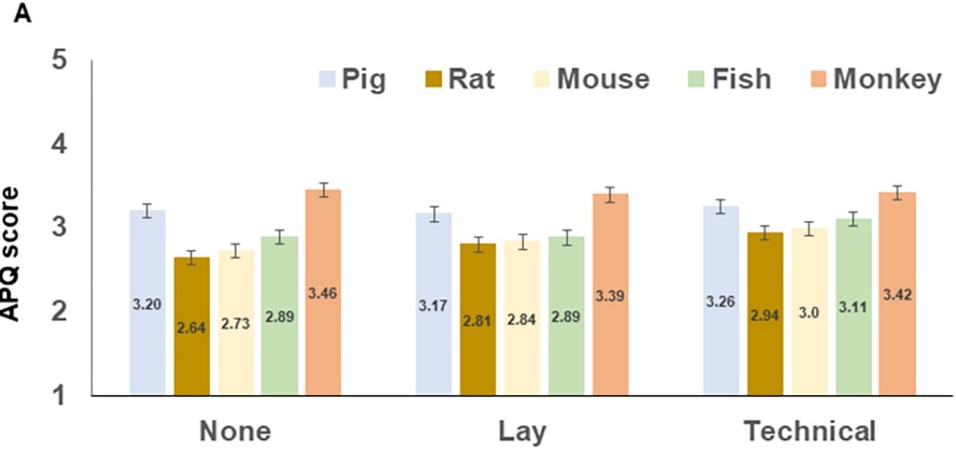

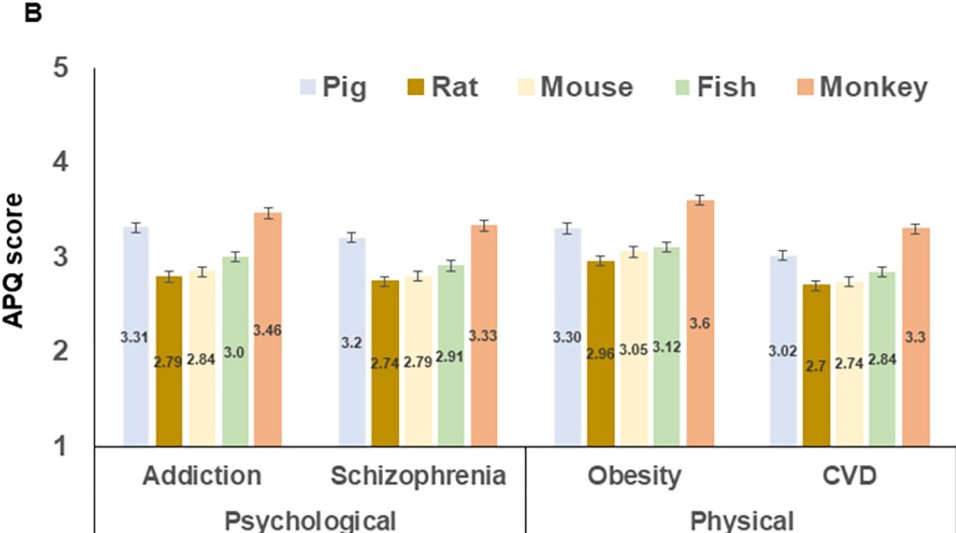

**Fig 2.** (**A**) Mean APQ scores across the 5 species (pig, rat, mice, fish and monkey) by survey (no summary, lay summary, technical summary). A score of 3 represents a 'neutral' response, with a score higher than 3 representing disagreement with the use of the species and a score lower than 3 meaning some level of agreement with the use of the species in research. The majority of differences between species were significant; error bars show the estimated standard errors for approximate visual comparisons. (**B**) Mean APQ scores across the 5 species (pig, rat, mice, fish and monkey) by the 4 research purposes (addiction, schizophrenia, obesity and cardiovascular disease, CVD). A score of 3 represents a 'neutral' response, with a score higher than 3 representing disagreement with the use of the species and a score lower than 3 meaning some level of agreement with the use of the species in research. Bars are split into psychological (addiction and schizophrenia) and physical disorders (obesity and CVD). For the statistical analyses, obesity and addiction were coded controllable, and schizophrenia and CVD were coded as uncontrollable. The majority of differences between species were significant; error bars show the estimated standard errors for approximate visual comparisons. In the middle of the bars are the data labels showing the means to two decimal places.

There was also an interaction between species and psychological (vs physical) disorder, $F_{(4,3 = 2095)} = 17.895$, $p < 0.001$, $\eta_p^2 = 0.029$. This arose because there were higher levels of disagreement with the use of rats ($p = 0.005$), mice ($p < 0.001$), pigs ($p < 0.001$), and monkeys ($p = 0.015$), in psychological as opposed to physical disorder research, whereas there was no difference in the case of fish ($p = 0.291$).

In the case of research into addiction (categorised as psychological and controllable), participants' attitudes towards using the species examined were different from each other, $p < 0.001$,

with the exception of attitudes towards the use of rats and mice, $p = 0.051$. In the case of research into schizophrenia (categorised as psychological but not controllable), participants' attitudes towards using the species examined were significantly different from each other, with the exception of attitudes towards the use of rats and mice, $p = 0.092$. In the case of research into obesity (categorised as physical and controllable), participants' attitudes towards using the species examined were all significantly different from each other, $p < 0.001$, with the exception of attitudes towards the use of mice and fish, $p = 1.000$. In the case of research into CVD (categorised as physical and uncontrollable), participants' attitudes towards using the species examined were significantly different from each other, with the exception of attitudes towards the use of rats and mice, $p = 0.279$. For full details of the pairwise comparisons, please see S3 Appendix Pairwise Comparisons (B).

Thus, although there were some differences by the category of research purpose, the numeric differences are small and the profile of differences by species are generally similar–participants typically differentiated across all species with the exception of rats and mice, or mice and fish in the case of obesity research.

Cutting across effects of species, there was a main effect of controllability, $F(1,595) = 81.204$, $p < 0.001$, $\eta_p^2 = 0.120$, and an interaction between controllability and psychological (vs physical) disorder, $F(1,595) = 47.480$, $p < 0.001$, $\eta_p^2 = 0.074$. The three-way interaction between controllability, psychological (vs physical) disorder and summary was also significant, $F(2,595) = 4.532$, $p = 0.011$, $\eta_p^2 = 0.015$.

Fig 3 shows how disagreement with the use of animals for controllable (addiction and obesity) versus non-controllable disorders (schizophrenia and CVD) depended on their psychological versus physical categorisation. The difference in APQ ratings for more versus less

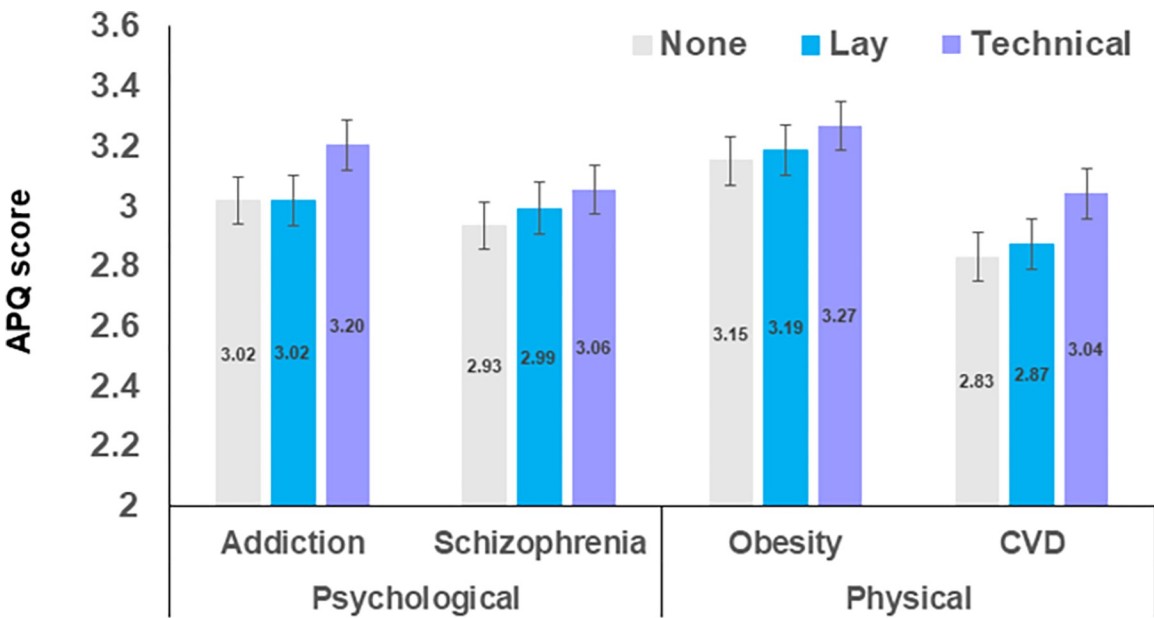

**Fig 3. Mean APQ scores collapsed across species by the 4 research purposes (addiction, schizophrenia, obesity and cardiovascular disease, CVD) by survey (no summary, lay summary, technical summary), rescaled to show small differences.** A score of 3 represents a 'neutral' response, with a score higher than 3 representing disagreement with the use of the species and a score lower than 3 meaning some level of agreement with the use of the species in research. Bars are split into psychological (addiction and schizophrenia) and physical disorders (obesity and CVD). There was more disagreement with the use of animals for obesity than for CVD research. There was also generally more disagreement with the use of animals for addiction than for schizophrenia research, but this difference was not seen in the no summary group; error bars show the estimated standard errors for approximate visual comparisons. In the middle of the bars are the data labels showing the means to two decimal places.

controllable psychological disorders, $p < 0.001$, for addiction and schizophrenia in the technical summary group, $p = 0.017$, and for addiction and schizophrenia in the no summary group, was not seen for addiction and schizophrenia after reading the lay summary, $p = 0.510$. As expected, research into obesity was associated with higher levels of disagreement with the usage, as compared to research into CVD, and in all of the summary groups, all $p < 0.001$.

## APQ profiles by eating orientation, gender and degree level education across summary type groups

Effects of categorically coded demographic factors by summary group were examined in a series of factorial analyses of APQ scores: firstly, because the known effects of these demographic factors could in principle confound differences otherwise attributed to summary group; secondly, to examine the role of demographic differences in mediating prejudice based on species, and the stigma attached to the disorder examined by way of research purpose.

There was a main effect of eating orientation, $F(1,537) = 18.174$, $p < 0.001$, $\eta_p^2 = 0.033$, because non-meat eaters showed overall higher levels of disagreement with animal use (Table 3). However, there were no significant interactions involving eating orientation, maximum $F(1,537) = 2.651$, $p = 0.104$, $\eta_p^2 = 0.005$, for the interaction with controllability.

There was both a main effect of gender, $F(1,555) = 30.789$, $p < 0.001$, $\eta_p^2 = 0.053$, and an interaction between species and gender, $F(3,1513) = 2.628$, $p = 0.033$, $\eta_p^2 = 0.005$. As shown in Table 3, females showed generally higher levels of disagreement with the use of animals in research, with distinctions drawn between each of the species sampled. Males showed somewhat lower levels of disagreement with the use of animals in research, and they did not differentiate the use of rats and fish, $p = 0.947$, or mice and fish, $p = 1.000$, though all other differences were significant. For full details of the pairwise comparisons, please see S3 Appendix Pairwise Comparisons (C).

The interaction between summary and gender, $F(2,555) = 3.845$, $p = 0.022$, $\eta_p^2 = 0.014$, arose because the usual sex difference (females showing overall reduced agreement with animal use, consistent with relatively higher pro-welfare tendencies) was seen in the no summary ($p < 0.001$) and lay summary ($p < 0.001$) but not the technical summary ($p = 0.246$) summary group. The ratings given by the males were more similar to the female profile in the technical summary group (Table 3). There were no interactions between summary and relevant degree level education, or eating orientation, maximum $F(2,567) = 2.531$, $p = 0.080$, $\eta_p^2 = 0.009$.

The three-way interaction between gender, controllability and psychological (vs physical) disorder was also significant, $F(1,555) = 5.635$, $p = 0.018$, $\eta_p^2 = 0.010$. This arose because male attitude ratings did not distinguish addiction and schizophrenia research, $p = 0.074$, whereas males showed higher disagreement with the use of animals for research into obesity than CVD, $p < 0.001$. Females showed higher disagreement with the use of animals for addiction

**Table 3. Mean APQ item ratings by survey and demographic factors.**

| Demographics | No summary | Lay summary | Technical summary |
|---|---|---|---|
| female (N = 383) | 3.148 (± 0.084) | 3.239 (± 0.096) | 3.205 (± 0.106) |
| male (N = 178) | 2.480 (± 0.159) | 2.436 (± 0.149) | 3.021 (± 0.118) |
| no relevant degree level education (N = 369) | 3.062 (± 0.101) | 3.005 (± 0.104) | 3.243 (± 0.092) |
| relevant degree level education (N = 204) | 2.918 (± 0.115) | 3.052 (± 0.131) | 2.812 (± 0.165) |
| meat eating (N = 467) | 2.915 (± 0.084) | 2.895 (± 0.089) | 3.070 (± 0.089) |
| non-meat eating (N = 76) | 3.353 (± 0.203) | 3.632 (± 0.218) | 3.630 (± 0.232) |

*Note*: Mean APQ item ratings (+ SEM) averaged across species and purpose) by survey summary group and demographic factors.

than for schizophrenia research, $p < 0.001$, and for research into obesity as compared to CVD, $p < 0.001$.

There was no main effect of relevant degree level education, $F(2,567) = 3.186$, $p = 0.075$, but there was an interaction between species and degree level education, $F(3,1546) = 5.341$, $p = 0.002$, $\eta_p^2 = 0.009$. Moreover, there was an interaction between species, degree level education, and psychological (vs physical) disorder, $F(4,1986) = 2.611$, $p = 0.041$, $\eta_p^2 = 0.005$. Those without a relevant degree did not differentiate the use of mice and fish, for both psychological ($p = 0.955$) and physical disorders ($p = 1.000$). Those with a relevant degree did not differentiate the use of mice and fish for research into physical disorders ($p = 0.659$) or the use of rats and mice, for both psychological ($p = 1.000$) and physical disorders ($p = 0.856$). They did differentiate the use of pigs and monkeys for physical ($p < 0.001$), but not for psychological research ($p = 0.091$). For full details of the pairwise comparisons, please see S3 Appendix Pairwise Comparisons (D). There were no other interactions by degree level education, maximum $F(1,567) = 2.765$, $p = 0.097$, $\eta_p^2 = 0.005$, for the interaction with controllability and psychological (vs physical) disorder.

Thus, as expected, there were effects of eating orientation, gender and relevant degree level education (discussed below). However, differences in the demographics of the summary groups do not explain the apparent effect of reading the technical summary. As shown in Table 1, there was a higher proportion of males in the technical summary group, but this difference does not confound interpretation of the overall differences seen by summary group, because the male responses seen in the technical summary group were closer to the female profile. This rather suggests that the intervention provided by the technical summary had a differentially greater effect in the males. Eating orientation and relevant degree level education did not interact with summary.

## Discussion

The APQ ratings provided the mechanism to distinguish attitudes to animal use by the five selected species and four biomedical research purposes, which were the focus of the study. As expected, the APQ ratings were correlated with more general measures of pro-welfare attitudes. With respect to the primary aim of the study, there was some evidence that providing participants with information about a research project summary influenced attitudes to the use of animals in such research. The no summary group provided a baseline measure of attitudes to the use of different species of animal for different medical research purposes, and hence a useful comparison for examining any effects of summary. However, neither the lay nor the technical summary produced any overall increase in agreement with animal use, and differences in the profiles of responses of participants provided with the lay versus the technical summary of the same research project were subtle, and not as expected in relation to species.

The findings from the APQ were consistent with overall effects of stigma, specifically related to distinctions based on controllability and the psychological versus physical nature of the disorders examined. There was overall less support for the use of animals for research into disorders viewed as controllable (addiction and obesity) rather than uncontrollable (schizophrenia and CVD), and this difference was accentuated for the physical disorders (obesity and CVD). However, the effects of the additional information provided by the summaries were again small and not as expected in relation to the nature of the disorder targeted by the research.

### Effects of species

As expected, there were clear overall differences in the levels of (dis)agreement with use of animals across the selected species, with the highest levels of disagreement with the use of pigs

and monkeys. These differences are likely related to the phylogenetic position of these species with respect to humans [37]. Disagreement with the use of pigs may also be attributed to the fact that pigs are familiar as domesticated animals and are known to show a wide range of emotions [38]. The progressively lower levels of disagreement with the use of mice, rats and fish were also as expected [24, 26]. Levels of agreement with the use of rats and mice were generally similar, consistent with their closeness in phylogenetic relations [39].

In the present study, the statistical interaction between summary group and species arose because after reading the research summaries, participants showed less differentiation between rats, mice and fish (and as much concern for fish as pigs after reading the technical summary). Counter to expectation, there was no selective effect on (dis)agreement with the use of rats, which were the subjects of the research described in the summaries.

### Effects of the disorder researched

The APQ was also used to test the prediction that perceived controllability of addiction and obesity should reduce levels of agreement with the use of animals for these specific research purposes (relative to schizophrenia and CVD). As predicted, perceived control over the disorder in question had a significant effect on levels of (dis)agreement with the use of animals for more stigmatised disorders [14, 18]. Participants overall agreed less with the use of animals in research into obesity compared to CVD, as well as for addiction as compared to schizophrenia research. Thus, the results confirmed the expected overall effect of controllability (perceived blameworthiness).

It was also expected that there should be overall more support for the use of animals for research purposes viewed as physical as compared to psychological. This was partially confirmed in that the effects of the physical versus psychological categorisation of the disorders depended also on controllability. The perceived control of the disorder was found to have a relatively greater impact in physical disorder research: the difference in disagreement with the use of animals for obesity and CVD research was bigger than the difference in disagreement seen between the use of animals in addiction and schizophrenia research.

The post-questionnaire ('your views') ratings reflecting beliefs that disorders were controllable showed positive correlations with the corresponding APQ ratings for addiction and schizophrenia (averaged by disorder across species). This positive correlation means that higher blameworthiness, in association with reduced control, predicted increased disagreement with animal use for these disorders. There was no such pattern of association for obesity or CVD. Thus, participants' reflective ratings showed some relationship between perceived controllability and attitudes to animal use, for the psychological but not the physical disorders. These associations were independent of the presumptions of stigma based on controllability set up in the APQ (as they were seen for both addiction and schizophrenia, but not for obesity).

As expected in general terms, the effects of perceived controllability and the psychological (vs physical) nature of the disorder on attitudes to animal use were influenced by participants' exposure to the research summaries. However, the specific profile of this effect was not as expected. There was no evidence that exposure to the summaries reduced overall disagreement with psychological research. There was no evidence that exposure to the lay summary reduced overall disagreement with the use of animals for more blameworthy disorders such as addiction. Participants given the lay summary were as concerned about the use of animals for schizophrenia as for addiction research. APQ ratings otherwise indicated more concern for animals used for addiction research (and for obesity compared to CVD in all summary groups).

This finding partially confirms the prediction that exposure to a research summary (particularly the lay version intended to be more accessible) should reduce distinctions based on perceptions of the research purpose. However, the distinction between schizophrenia and addiction was reduced because of increased disagreement with the use of animals for schizophrenia research rather than reduced disagreement with the use of animals used for addiction research, which would be expected if improved understanding of animal research increased agreement with animal use.

## Blameworthiness and control

There have been initiatives to reduce negative stereotyping [10], and with some evidence of success [11]. However, various studies of the impact of mental health awareness campaigns on public attitudes have found that informational interventions have only small impacts on attitudes [40, 41], and only sometimes reduce stigmatisation [14].

The current study provides systematic evidence on how persistently negative assumptions that exist surrounding addiction and obesity can have wider effects, here on the (dis)agreement with the use of animals in research conducted in order to better understand such disorders. Presumptions of controllability are at odds with the evidence to suggest the importance of genetics as an underlying factor, based for example on estimates of heritability for addiction [42]. Similarly for obesity, studies show heritability of body mass index and obesity [16] as well as the role of specific genes [43, 44]. Moreover, although there are clear environmental influences on the development of addiction and obesity, there are also environmental risk factors influencing the more 'uncontrollable' illnesses, CVD and schizophrenia, including high fat diet and the use of certain addictive substances, respectively. Indeed, a number of the causes of CVD (e.g., smoking, being overweight, inactivity, excessive alcohol consumption) might also be thought to be controllable.

## Alternative measures of pro-welfare tendencies

Previous research has found stronger empathic concern to be linked with higher levels of sensitivity to use of animals in experimentation [45, 46]. In the present study, the correlational analyses confirmed this conclusion for the measure of empathy provided by the EQ. However, it should be noted that the correlations with the other pro-welfare indicators were of relatively low effect size (between .12 and .17), suggesting that empathy was not the most useful predictor of attitudes towards animal use.

Also consistent with previous research examining attitudes to animal use across a wider range of purposes, the present study showed a positive relationship between APQ and AAS scores [24, 26], and a greater belief in animal mind was associated with lower agreement with the use of animals in biomedical research across species, as measured by the AAS and APQ totals [24].

## Effects of demographic factors

We have previously found a weak positive correlation between age and APQ attitudes [27], but this finding has been inconsistently reported in studies of predominantly younger adults [24, 26]. No relationship between age and any of the measures of pro-welfare tendencies was found in the present study. However, there was a restricted range of ages: the majority of participants were young adults, and with a relatively young average age of around 30 years. In any case, there have been mixed findings in reports of the effects of age on attitudes to animal use, and age confounds cohort effects [2].

In line with previous studies using the APQ [24, 26, 27], vegetarians and vegans showed overall higher disagreement with animal use than did meat eaters. This effect did not depend on species, or the nature of the disorder investigated.

Females show generally higher levels of disagreement with the use of animals in research [2, 24, 26, 27]. In the present study, females drew distinctions between each of the species sampled, whereas males did not consistently differentiate the use of rats, mice and fish. With respect to the research purposes examined, whilst females showed higher disagreement with the use of animals for addiction as compared to schizophrenia research, and for research into obesity compared to CVD, male attitude ratings did not distinguish addiction and schizophrenia. However, like females, males showed higher disagreement with the use of animals for research into obesity relative to CVD. Thus, the overall higher levels of disagreement in the female participants were accompanied by finer distinctions based on species and purpose of use.

Previous research has also shown the link between greater support for animal use in research and higher education [47]. In the present study, there was no overall difference between APQ scores for those with relevant degree experience, but there was some effect dependent on species. Participants reporting a relevant degree differentiated most species pairings except for rats and mice. Participants without any relevant degree showed a slightly different profile in that they did not distinguish the use of mice and fish, but all other species pairings were differentiated. Thus, given the generally similar ratings for the use of rats, mice and fish, and the small numeric difference in the ratings, the effect of degree level education can be seen as limited. This finding, combined with the limited effects of the summaries on APQ ratings in relation to disorder, would seem to suggest that education or knowledge does not necessarily influence attitudes, despite previous research suggesting otherwise [1, 10, 11]. Our results are more in line with findings that scientific knowledge, or lack thereof, does not have a consistent relationship with attitudes towards animal research [48].

## Limitations

The five species included in the current research were only a small selection of those used in biomedical research, for example, insects are being used more frequently [49]. Similarly, with respect to purpose, further studies would be necessary to test the generality of findings using a wider range of disorders, for example, depression and anxiety are highly prevalent and yet still potentially stigmatised [9].

Participants' perceptions of the controllability of the selected disorders were presumed based on previous studies and had not been piloted for the current study. Perceptions of controllability were measured after completion of the questionnaires, by participants' reflective ratings. However, the observed patterns of association were independent of the presumptions of stigma based on controllability set up in the APQ (as they were seen for both addiction and schizophrenia, but not for obesity). Moreover, as discussed above, there are controllable environmental risk factors influencing the more 'uncontrollable' illnesses, i.e., CVD and schizophrenia.

We acknowledge that the selected lay summary may not have been as generally accessible as intended and the corresponding technical summary may have introduced some bias by seeming to legitimise such research for participants 'blinded by science' in the absence of understanding. However, we saw higher APQ scores, reflecting greater disagreement (and more pro-welfare attitudes) in the case of fish, following the presentation of the technical summary (not different from attitudes towards the use of pigs), and there was no specific mention of fish in the technical summary. Moreover, in both summary groups the direction of change in APQ

scores reflected increased disagreement with the use of smaller animals. Further research in this area should assess the efficacy of different types of lay summaries, written according to different guidelines [50] and protocols for co-creation with stakeholders [51].

Distribution of the survey links was intended to reach participants with matched demographics, and we analysed to see how far the sampling strategy had been successful. There was nothing to suggest that the observed differences between the survey groups might explain the observed pattern of results. It must nonetheless be acknowledged that there was differential drop-out from the different survey groups. Drop-out rates for the no summary and lay summary groups were similar and low (15.1% and 17.6%, respectively). Drop-out for the survey preceded by the technical summary was much higher (37.2%). This difference could suggest that the samples reached were not well matched in terms of their motivation to complete the survey. Alternatively, the differential drop-out rates may be evidence that the participants were disengaged by the technical summary. Irrespectively, to set up randomisation to survey allocation within the Qualtrics would have been a better sampling strategy.

As per previous studies, the age range was limited, and the majority of participants were female [24, 26]. However, age was well-matched across the summary groups, and the total sample of males studied was not small (N = 178). More males were in the technical summary group, and a smaller proportion of those completing the survey reported relevant degree-level education. Participants who read the lay and technical summaries had similar views on the justification of research for psychological relative to physical illnesses. They reported similar experience of the conditions mentioned in their friends and/or family. They rated the health risks of addiction and schizophrenia as similarly beyond an individual's control. However, participants who read the technical summary reported higher levels of agreement with the 'your views' statement that the health risks of obesity and CVD are beyond the individual's control. Thus, although the participants completing the different survey variants were generally well-matched, there were some differences.

Finally, it must be acknowledged that whilst a large sample has its advantages, effects showing as significant can be of small effect size, limiting the implications of the findings. Notably, the effects of exposure to the research summaries, demographic factors and participants' self-reported views were all of statistically small effect size. In line with other studies with the APQ [24, 26, 27], the effect of species was relatively large $(\eta_\mathrm{p}^2 = 0.180)$. The present study examined differences in attitudes to animal use within a narrow range of medical research purposes and by two factors—psychological (vs physical) nature of disorder and the controllability of disorder–and the majority of the effects pertaining to purpose were small to medium (largest $\eta_\mathrm{p}^2 = 0.120$, for controllability).

## Conclusions and implications

Participants given the lay summary were as concerned about the use of animals for schizophrenia as for addiction research. APQ ratings otherwise indicated more concern for animals used for addiction research (and for obesity than CVD in all summary groups). Thus, there was no evidence to support the assumption that the lay summary should increase agreement with animal use.

The findings of the present study show differences in attitudes to animal use based on both prejudices (based on perceived control), additional information (in the form of exposure to a research project summary) and species. The observed effects of stigma can be seen to relate to commonly reported misconceptions about addiction and obesity [13, 14, 15, 52]. However, there was no evidence for any particular benefit of the lay summary and some of the effects of the research summaries were not in line with predictions. The findings of the present study

suggest that improved understanding of the factors influencing attitudes to animal use may be essential to improve the effectiveness of the communication activities of biomedical scientists. For example, information on vulnerability to disorders seen as blameworthy or psychological may help to mitigate the effects of perceived controllability on public agreement with animal use for translational research.

## Supporting information

**S1 Appendix. The research project summaries.**
(DOCX)

**S2 Appendix. All data.**
(SAV)

**S3 Appendix. Simple main effects pairwise comparisons.**
(DOCX)

## Author Contributions

**Conceptualization:** Helen J. Cassaday, Lucy Cavenagh, Hiruni Aluthgamage, Aoife Crooks, Charlotte Muir.

**Data curation:** Helen J. Cassaday, Lucy Cavenagh, Hiruni Aluthgamage, Aoife Crooks.

**Formal analysis:** Helen J. Cassaday, Lucy Cavenagh, Hiruni Aluthgamage, Aoife Crooks.

**Funding acquisition:** Helen J. Cassaday, Charlotte Bonardi, Carl W. Stevenson.

**Investigation:** Helen J. Cassaday, Lucy Cavenagh, Hiruni Aluthgamage, Aoife Crooks, Lauren Waite, Charlotte Muir.

**Methodology:** Helen J. Cassaday, Lucy Cavenagh, Hiruni Aluthgamage, Aoife Crooks, Lauren Waite, Charlotte Muir.

**Project administration:** Helen J. Cassaday, Charlotte Bonardi, Carl W. Stevenson.

**Resources:** Helen J. Cassaday.

**Supervision:** Helen J. Cassaday.

**Writing – original draft:** Helen J. Cassaday, Lucy Cavenagh, Hiruni Aluthgamage, Aoife Crooks.

**Writing – review & editing:** Helen J. Cassaday, Charlotte Bonardi, Carl W. Stevenson, Lauren Waite, Charlotte Muir.

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
