## [Decision Letter · Decision Letter 0]

17 May 2023

PONE-D-23-08435Attitudes to the use of animals in biomedical research: Effects of stigma and selected research project summariesPLOS ONE

Dear Dr. Cassaday,

Thank you for submitting your manuscript to PLOS ONE. After careful consideration, we feel that it has merit but does not fully meet PLOS ONE’s publication criteria as it currently stands. Therefore, we invite you to submit a revised version of the manuscript that addresses the points raised during the review process.

The manuscript has been read by 2 reviewers and myself. The reviewers and I are largely in agreement that the paper represents potentially valuable data on an important topic. However, they also feel that quite some revision is required, particularly around the aims, rationale, and hypotheses. Again, I concur.

On my reading, the first paragraph does not adequately introduce the aim. Really, you are examining the effect of summaries *in the context* of attributes of health conditions that are relevant to lay beliefs, or indeed to stigma (such as controllability etc). In fact, a finding that people object more to animal testing for controllable illnesses would be indirect evidence of structural stigma in that regard. However, this key aspect of the aim is not really introduced at all and, consequently, the material on controllability etc just comes out of the blue. "Why these factors?", we might ask. I would like you to add a sentence or two of rationale in the first paragraph, as you have done for the idea of lay summaries in general. This would situate the rest of the Introduction much more clearly for the reader. I suggest you make this change and try to structure the introduction more clearly around it, rather than using the methods to introduce variables.

I also agree with Reviewer 1's specific point that the hypotheses do not reflect the complexity of the design. Although I well understand the (wise) decision to avoid the 4-way as it was not of theoretical interest, I would like you to consider the possible 2-way (maybe even 3-way) effects that would flow from your aims and design. Please note that Reviewer 1 has provided very detailed comments in a separate file and these need to be addressed. Finally, Reviewer 2  makes a series of very good suggestions that I recommend you address.

Please address these issues and resubmit your manuscript following the guidelines in this email, and I would be glad to consider it again.

We look forward to receiving your revised manuscript.

Kind regards,

Stefano Occhipinti

Academic Editor

PLOS ONE

Journal Requirements:

"This work was in part supported by the Biotechnology and Biological Sciences Research Council [grant number BB/S000119/1]."

"This work was in part supported by the Biotechnology and Biological Sciences Research Council  https://www.ukri.org/councils/bbsrc/ [grant number BB/S000119/1] awarded to HJC, CWS and CB.

"This study examines the effectiveness of project summaries outlining a plan of work supported by the Biotechnology and Biological Sciences Research Council [grant number BB/S000119/1] as moderators of attitudes to animal use. The BBSRC had no further role in the study. The authors have declared that no individual competing interests exist."

Reviewers' comments:

Reviewer's Responses to Questions

**Comments to the Author**

1. Is the manuscript technically sound, and do the data support the conclusions?

Reviewer #1: Yes

Reviewer #2: Yes

2. Has the statistical analysis been performed appropriately and rigorously? 

Reviewer #1: Yes

Reviewer #2: Yes

3. Have the authors made all data underlying the findings in their manuscript fully available?

Reviewer #1: Yes

Reviewer #2: No

4. Is the manuscript presented in an intelligible fashion and written in standard English?

Reviewer #1: Yes

Reviewer #2: Yes

5. Review Comments to the Author

Reviewer #1: This manuscript examines attitudes towards the use of animals in biomedical research, specifically the effects of the reading research summaries (none, lay, technical) on expressed attitudes towards animals in biomedical research. The effect of the controllability of the disorder being researched was varied, and consistency across different animal species was also examined. The topic is a very interesting one, and the study has a lot of potential. As it currently stands it needs quite a bit of editing, particularly in the Introduction, Methods, and Discussion sections to more clearly convey the rationale for the study and why the work is important. In addition, the hypotheses as currently stated are simple, but the design is a 4-way mixed design; the hypotheses should reflect this complexity. There are also methodological aspects of the study that need to be clarified, including whether participants were randomised to the between-subjects conditions and whether any of the materials were piloted beforehand.

Reviewer #2: This paper provides valuable data on the utility of summaries for scientific research in relation to the acceptability of animal use with two clear findings. First, having a summary whether technical or lay increase the acceptability of the use of animals for some species. Second, not all use of animals for medical research are equal with some diseases like obesity or addiction showing less favourable results for the use of animals compared to CVD.

General comments the manuscript could benefit from a clear presentation of the aim and specific hypotheses in the introduction to help provide structure to the results section and also to help provide an outline of the paper for the readers so the results section feels a little more structured. Second, much clearer description of IV and DV required in the methods section and should not be held till the beginning of the analyses. Clear statement of precisely which type of analyses is being conducted in each of the different sections of the results and also a thorough check on the formatting of all test strings reported. The paper does provide some interesting insights into the use of animals and raises interesting ethical issues around use of animals around perceived deserving/undeserving conditions.

Specific Comments:

Page 4. Psychoeducation interventions (…. and then no closing bracket.

Why was the drop out rate for completing the third survey so high compared to the other two sureys (i.e. loss of 119 participants so almost a third of the starting sample)? Consider using missing data techniques like multiple imputation to reduce bias and increase sample size.

Measures of reliability like Cronbach alpha would be good to see in the material section of the questionnaire. It is a little odd that they are not here as even though these scales are relatively well known it is still valuable information to report.

For the EQ questionnaire what is the justification for the categorisation of a continuous variable? Please provide appropriate citations or use as continuous variable.

Please include summary descriptive statistics in sample section to give readers a sense of gender, eating orientation etc as we know these factors impact evaluative judgements around the acceptability of animal use.

Clear explanation of what the repeated measures were would be helpful i.e. did the participant complete each of the questionnaires multiple times (if so how many times) and evaluating what each time (presumably different types of animals to be used). In the methods there is no mention of the species used nor the perceived control/blame factor nor the different types of diseases. Please re-write this section with all between and repeated measure subjects clearly specified. Readers should not have to wait till the top of the analysis section to find this out.

Results mention tests for sphericity but not tests for normality of residuals and heterogeneity/homogeneity could these be included.

In the section entitled Overall difference between summary variants…. The chi square results are reported in an odd fashion. Please re-write as (2,561) = 24.62, p ….. Same for the rest of chi-square analysies.

In general test string could be reported up to APA standard with appropriate spacings and italics being used i.e. on p for example. Please amend throughout manuscript.

Clear description of multivariate test showing technical summary reported higher disagreement that the health risk of obesity are beyond an individual control compared to what? The lay and no summary groups? Please make this clearer.

Bottom of page 17 analyses between addiction and species use should point readers to a table either in text or in supplementaries where readers can view the actual test strings as opposed to relying on general statements i.e. all less than .001.

Please can you provide clearer link to the data, analysis script and all materials used in the manuscript preferably one that is held in some repository like the OSF.

6. PLOS authors have the option to publish the peer review history of their article (what does this mean?). If published, this will include your full peer review and any attached files.

Reviewer #1: No

Reviewer #2: No

---

## [Author Response · Author response to Decision Letter 0]

19 Jun 2023

Responses to reviewers

We thank the Editor and Reviewers for their constructive comments to improve the manuscript. Responses to Reviewer #1’s comments provided in the separate file are included below. The points raised are addressed below and revisions to the manuscript are highlighted. Our own minor corrections are also highlighted.

Editor

On my reading, the first paragraph does not adequately introduce the aim. Really, you are examining the effect of summaries *in the context* of attributes of health conditions that are relevant to lay beliefs, or indeed to stigma (such as controllability etc). In fact, a finding that people object more to animal testing for controllable illnesses would be indirect evidence of structural stigma in that regard. However, this key aspect of the aim is not really introduced at all and, consequently, the material on controllability etc just comes out of the blue. "Why these factors?", we might ask. I would like you to add a sentence or two of rationale in the first paragraph, as you have done for the idea of lay summaries in general. This would situate the rest of the Introduction much more clearly for the reader. I suggest you make this change and try to structure the introduction more clearly around it, rather than using the methods to introduce variables.

I also agree with Reviewer 1's specific point that the hypotheses do not reflect the complexity of the design. Although I well understand the (wise) decision to avoid the 4-way as it was not of theoretical interest, I would like you to consider the possible 2-way (maybe even 3-way) effects that would flow from your aims and design. 

Response: We have made the suggested changes to clarify the aims, rationale, and hypotheses (in relation to the factorial design). We have also used the term controllability more consistently throughout, keeping the presumed relationship with blameworthiness more for the Discussion (though it is signposted earlier).

Reviewer #1

As it currently stands it needs quite a bit of editing, particularly in the Introduction, Methods, and Discussion sections to more clearly convey the rationale for the study and why the work is important. In addition, the hypotheses as currently stated are simple, but the design is a 4-way mixed design; the hypotheses should reflect this complexity. 

Response: We have made the suggested changes to clarify the aims, rationale, and hypotheses (in line with the Editor’s feedback above), p4-5 and p7-8. We have also elaborated in the Discussion (e.g. p25-26) and clarified the importance/implications p35. Changes in the Methods section are also highlighted.

There are also methodological aspects of the study that need to be clarified, including whether participants were randomised to the between-subjects conditions and whether any of the materials were piloted beforehand.

Response: Participants were not randomised to the survey groups, distribution of the survey links was intended to reach participants with matched demographics (p10; Table 1) and we analysed to see how far this matching had been successful (reported p18-19). There were some differences, but eating orientation was not different by summary group allocation, the survey groups were well matched for age, and the overall levels of pro-welfare tendencies (as measured by the total scores for the APQ, AAS, BAM and the EQ) were similar across the summary groups. As explained p25, the unexpected higher proportion of males in the technical summary group does not explain the observed pattern of results.

To set up randomisation within the Qualtrics would have been better and this design limitation is acknowledged p33-34 in the revision.

The materials were not piloted for this specific study as such, but we have worked with different APQ variants in a number of previous studies (three cited in the text) and the other questionnaires used are well-established. The research project summaries were taken from our UKRI-funded project (provided in the S1 Appendix; they were edited only to match them for length).

Abstract

• Needs to be edited for clarity – some info about methods shared mid-way through the abstract, making it harder to differentiate what is methods/results. The abstract doesn’t make it clear how stigma is related to the research that was conducted 

• Were the summaries presented between-groups? What about disorder types and animal species? Edit to make this clearer

• The final sentence about findings could be more impactful – instead of saying “found effects of summary” comment on whether type of information about a study affects attitudes to use of animals in biomedical research. 

Response: The Abstract has been edited and clarified in line with these suggestions.

Introduction

• Pg 3 – is it barriers to general public/lay understanding being examined? Suggest state explicitly 

Response: The text has been clarified p4.

• Pg 4 – paragraph on obesity stands out on its own, unclear how it links to the overarching discussion of stigma and mental illness; this section needs an overarching sentence(s) to tie together the relevance of stigma to the study aims – this link wasn’t clear to me from what is written. Having read more of the paper I now know that you will be using obesity as one of the conditions – the reader doesn’t know this yet in the intro, so the relevance needs to be explicitly stated/worked into the thread of the argument

Response: The linking has been improved by the introduction of the study aims (as suggested by the Editor p4-5) and the link to stigma has been made explicit p6. 

• Animal species section needs to be tied back in to overarching study aims

Response: This has been done p7.

• Not clear to me from what is written how stigma is related to the effect of different ways of describing studies to lay people and how that relates to attitudes towards biomedical research. This needs to be explicit 

Response: The study aims have been made explicit p4-5.

• There isn’t much in the introduction about what is currently known about attitudes. While there may not be much research specifically looking at how information is presented (e.g., in summaries) and how that impacts attitudes, there is work looking at attitudes to biomedical research more generally that could be discussed

Response: We have cited some additional papers to cover the wider question of attitudes to biomedical research, including a review, p4 and p29.

• Section “Measuring effects of selected research project summaries” (p 5) reads like a methods section. I’m not certain this section contributes to the overall argument of the Introduction

Response: This section (and much of its content) has been removed.

• Aims and Hypotheses- too much of this section reads as a methods section. This needs to be more about the constructs, rather than describing what was done.

Response: The description of the methods as such has been reduced p7 and the hypotheses have been elaborated p8-9.

• The 3 predictions should be more specific, and include comparison groups. Phrases like “should reduce”, “should be relatively more support” need to be tightened up. Having read further in the paper, the hypotheses outlined in the Introduction are very simple compared to the very complex 3x5x2x2 design outlined on p14. The hypotheses need re-writing to match the complexity of the study design and expected findings

Response: We have tightened up the general predictions and elaborated on the hypotheses p8-9. 

Methods

• How were participants allocated to the summary groups? Was it random? When you say “different versions of the survey were not available at the same time” does this refer to the different summary groups?

Response: Participants were not randomised to the survey groups. Different versions of the survey for the different summary groups were not made available to the same sample of participants. The text has been clarified p10. 

Distribution of the survey links was intended to reach participants with matched demographics and we analysed to see how far this matching had been successful (reported p18-19). There were some differences, but eating orientation was not different by summary group allocation, the survey groups were well matched for age, and the overall levels of pro-welfare tendencies (as measured by the total scores for the APQ, AAS, BAM and the EQ) were similar across the summary groups. As explained p25, the unexpected higher proportion of males in the technical summary group does not explain the observed pattern of results.

To set up randomisation within the Qualtrics would have been better and this design limitation is acknowledged p33-34 in the revision.

• Unclear why ANOVAs are referred to in the participants section.

Response: The specific reference to ANOVA has been removed, the intention is to explain the exclusions as the context to the demographic data provided.

• Description in Materials needs to be more specific. For example, What is ‘eating preference’ referred to on p9? Is it the same as ‘eating orientation’ on p11. Suggest that you are specific about the number of questions rather than saying phrases such as “a few additional questions”

Response: In the revision we now consistently refer to ‘eating orientation’ (rather than interchangeably referring to ‘eating preference’). As suggested, we have been more specific about the 7 additional questions asked, p11-12.

• Measures – what is the range of possible scores for each measure

Response: The range of possible scores has been specified for each measure (e.g. 1-5 per item for the 5-point scales; multiplied by the number of items for the scale totals). 

• There is some repetition of information within the method

Response: The repetition of ethics information has been removed p16 (and we refer back to the additional details incorporated in the ethics statement, p9). We’ve left these additional details with the ethics, as required by the editorial manager in a pre-submission check.

• Information such as diseases designated controllable vs uncontrollable, and the physical vs psychological should be introduced prior to the ‘Data analysis’ section as it’s the design of the study

Response: The design information has been provided earlier p11.

• It is unclear whether the summaries were piloted – were they in fact lay vs technical in the view of the general public?

Response: The summaries were published and made openly available as lay versus technical research summaries (by a UK funding body). They’re taken from our funded project and the grant holders (including myself) are authors on this paper. The data were collected by the other authors.

The study presented here was intended to assess the effectiveness of these summaries in moderating attitudes to animal use without further editing (except the removal of some text from the lay summary, in order to match the length of the technical summary, and to present as a single paragraph as per the technical summary). The Methods have been clarified p12. The summaries and their source are provided in the S1 Appendix.

Results

• This study is trying to do a lot all in one, and as a result it’s difficult to determine what the key results are. The complexity of the study design means clarity of writing and presentation of information is extra important.

Response: We have acted on the specific suggestions made, these have been a great help and have improved the overall clarity. 

Headings need to be clear and be linked to the constructs being examined, not just the survey components. For example the heading “Overall differences between the summary variants of the surveys” seems to be describing differences in participant characteristics across the summary type conditions

Response: This heading has been revised and p19, and we have revised the other headings to better relate to the constructs being examined. 

• The clarity of results displayed in charts would be increased having numbers on top of the bars on the charts to show the means. The mean differences are not very large, so it would be helpful to see the specific numbers when interpreting results. 

Response: Numbers to show the means have been inserted, using the central option to avoid overlap with the error bars.

• Table 3 title is ‘Differences in ratings’ are these mean differences? It would be good to see the means themselves, not just differences. 

Response: Apologies, these are mean item ratings (showing differences by survey summary group, but not difference scores). The table title and legend have been clarified.

• When interactions were found to be non-significant were they removed from the model and the analyses re-run to examine the relationships of interest?

Response: There was no intentional removal of interactions. We ran the standard SPSS mixed design repeated measures model. To my knowledge SPSS does not allow the removal of non-significant interactions so this does not need stating in the text. In the process of re-generating the pairwise comparisons for fuller reporting, as required by Reviewer #2, all the ANOVAs were re-checked. We did identify one analysis conducted with a missing factor. We had previously run the analyses for the effects of degree level education without the survey summary factor. This was unintentional and has been corrected in the revision p24, so (as described in the Data analyses section) the analysis for the effects of degree level education matches those conducted for eating orientation and gender (and presented directly above this section). Changes are highlighted, the main conclusions are unaltered.

Discussion

This section needs harmonising with the introduction. 

Response: The Introduction has been improved, to provide a better frame for the hypotheses examined. The Discussion now summarises how the findings relate to the general predictions to be tested (p25-26) before going on to a more detailed consideration of the findings, as per the headed sections. We now explicitly consider the implications of the findings at the end of the Discussion, p35. 

• Some parts read more as a summary of results than an interpretation with links to existing research (for example, p23, paragraph 3)

Response: This paragraph has been improved by removal of the results-style sentence. We have checked for other sentences of this kind and added additional links to existing research (e.g., p30-31). We have retained some of the summary statements where we feel these are necessary, to help the reader keep track (as it is a complicated study).

• The discussion of CVD as potentially being perceived as controllable (p25) seems to me to be something that should be included in the limitations section. Was controllability piloted? I don’t remember reading that it was. 

Response: Controllability was presumed based on previous studies. It was not piloted for the current study. This point is now discussed in the limitations section, p32.

• There was not as much discussion of the lay summary vs no summary vs technical summary differences as I would have expected. I would have thought that the no summary group gives you an idea of people’s general attitudes and a useful comparison for examining any effects of summary.

Response: Some further discussion of the no summary group has been provided p25-26.

Other

• Figure 1 appears to contain both an example of a survey question and study results?

Response: The survey format is now shown separately as the new Figure 1. This also helps to clarify how the questions are repeated systematically for each of the species.

• Figures 1 & 2 – lower bound of the CI?SE? isn’t visible on some of the colours (also suggest a note clarifying what the bars on the charts represent)

Response: The colours have been changed (and checked for greyscale discriminability) so the standard error bars are visible. The figure legends have been updated to clarify what the bars show.

Reviewer #2

… the manuscript could benefit from a clear presentation of the aim and specific hypotheses in the introduction to help provide structure to the results section and also to help provide an outline of the paper for the readers so the results section feels a little more structured. 

Response: The aims and hypotheses have been clarified p4-5 and p7-8.

Second, much clearer description of IV and DV required in the methods section and should not be held till the beginning of the analyses. 

Response: As suggested, the variables have now been described earlier p11 (inserted clarifications are highlighted).

Clear statement of precisely which type of analyses is being conducted in each of the different sections of the results and also a thorough check on the formatting of all test strings reported. 

Response: Throughout the Results, the test string formats have been made consistent as suggested below.

Specific Comments:

Page 4. Psychoeducation interventions (…. and then no closing bracket.

Response: This has been corrected.

Why was the drop out rate for completing the third survey so high compared to the other two sureys (i.e. loss of 119 participants so almost a third of the starting sample)? Consider using missing data techniques like multiple imputation to reduce bias and increase sample size.

Response: This is a good point, particularly as the drop out rates for the no summary and lay summary groups were so similar. The third survey was preceded by the technical summary. In the revised Discussion we acknowledge the limitation of the differential drop-out p33. However, this may be a real effect if participants were disengaged by the technical summary. If this is the correct interpretation of the differential drop out, multiple imputation could exaggerate the group differences. In any case, there is no suitable average with which to replace missing values because the individual differences in attitudes to animal use (by species and purpose) are of interest.

Measures of reliability like Cronbach alpha would be good to see in the material section of the questionnaire. It is a little odd that they are not here as even though these scales are relatively well known it is still valuable information to report.

Response: In the present study, the AAS showed good internal consistency across the 10 items (� = 0.774). The APQ showed excellent internal consistency across the 20 items for 5 species by 4 research purposes (� = 0.980). The BAM showed excellent internal consistency across the 20 items of four matched questions for each of the 5 species (� = 0.906).

These Cronbach’s alpha values were similar to those reported in other studies conducted using these scales and are now reported, for the AAS (p13), APQ (p13) and BAM (p14). The EQ uses a Likert scale but the scoring precludes the use of the Cronbach statistic. The test-retest reliability for the EQ has previously been reported to be very good (r = 0.97; Baron-Cohen & Wainwright, 2004), now reported p15.

For the EQ questionnaire what is the justification for the categorisation of a continuous variable? Please provide appropriate citations or use as continuous variable.

Response: We followed the EQ scoring reported by the authors of this scale (Baron-Cohen & Wainwright, 2004). This is clarified p14.

Please include summary descriptive statistics in sample section to give readers a sense of gender, eating orientation etc as we know these factors impact evaluative judgements around the acceptability of animal use.

Response: The suggested summary descriptive statistics are now provided in the sample section p10. Of the in total 598 participants who started the study, 178 participants identified as male and 383 participants identified as female. In total 467 participants identified as omnivore and 76 participants identified as non-meat eating (vegetarian or vegan). In total 204 participants reported a relevant degree and 369 participants reported no relevant degree. We have retained Table 1 to facilitate direct comparison of the participants’ demographics across the (none, lay, technical) survey summary types.

Clear explanation of what the repeated measures were would be helpful i.e. did the participant complete each of the questionnaires multiple times (if so how many times) and evaluating what each time (presumably different types of animals to be used). In the methods there is no mention of the species used nor the perceived control/blame factor nor the different types of diseases. Please re-write this section with all between and repeated measure subjects clearly specified. Readers should not have to wait till the top of the analysis section to find this out.

Response: The Methods have been clarified p11, also the revised Figure 1 which makes the APQ format explicit.

Results mention tests for sphericity but not tests for normality of residuals and heterogeneity/homogeneity could these be included.

Response: In the present study, we had relatively large N. Parametric tests assume normality of the distribution of the means and Central Limit Theorem shows that (even for sample sizes very much lower than the one used in the present study) the means will approximate a normal distribution, irrespective of distributions of the raw data (Myers & Well, 2003; Norman, 2010). Moreover, the data are comprised of responses over a number of items for each of the scales.

There was homogeneity of variances between groups as assessed by Levene’s test for equality of variance for the AAS, BAM and the EQ scores, p > 0.05. Homogeneity of variances was violated for the APQ total scores, p = 0.012. However, the survey groups had roughly equal sample sizes so Levine’s test is not strictly necessary.

Shapiro-Wilk tests of normality were failed, p < 0.05, for AAS, APQ, BAM and the EQ scores across the 3 summary groups, with two exceptions, for BAM total no summary (p= 0.051) and EQ total technical summary (p = 0.096). However, non-normality is less of a problem in large samples and the Type 1 error rate is relatively unaffected by non-normality (Myers & Well, 2003).

Parametric tests are more sensitive and powerful than their non-parametric alternatives (and there are no good non-parametric alternatives for a mixed-ANOVA). Parametric tests are also very robust, so even when their assumptions are violated there is low risk of drawing the incorrect conclusion (Myers & Well, 2003; Norman, 2010). 

Different considerations apply to correlational approaches which are more sensitive to individual data at the extremes of the distribution but which have nonetheless been shown to be robust in simulation studies (Havlicek & Petersen, 1976). 

In the revision, Shapiro-Wilk and Levene’s tests are now reported, provided together with justification of the parametric approaches adopted p17-18.

In the section entitled Overall difference between summary variants…. The chi square results are reported in an odd fashion. Please re-write as (2,561) = 24.62, p ….. Same for the rest of chi-square analysies.

Response: The chi square results have been re-written as suggested.

In general test string could be reported up to APA standard with appropriate spacings and italics being used i.e. on p for example. Please amend throughout manuscript.

Response: The test strings have been amended throughout. All the statistics have been re-checked and a couple of further Greenhouse-Geiser corrections have been applied. The statistical conclusions are unchanged, but the adjustments were omitted for some of the ANOVAs run with the demographic factors. All numeric changes are highlighted (just not the recurrent format amendments).

Clear description of multivariate test showing technical summary reported higher disagreement that the health risk of obesity are beyond an individual control compared to what? The lay and no summary groups? Please make this clearer.

Response: Participants in the technical summary group reported higher agreement that the health risks of obesity and CVD are beyond the individual’s control, as compared to the levels of agreement indicated by participants in the lay summary group. The text has been corrected and clarified p19.

Bottom of page 17 analyses between addiction and species use should point readers to a table either in text or in supplementaries where readers can view the actual test strings as opposed to relying on general statements i.e. all less than .001.

Response: We agree that the text description of the test details would be very dense given the number of (Bonferroni-corrected) comparisons. Please see S3 Appendix for details where the manuscript text has not been elaborated to provide actual test results. These test details are supplied in the suggested table format (adapted from the SPSS outputs) labelled A-D and referred to in the text. An error in identifying the largest significant p has been corrected (also highlighted).

We previously ran the analyses for the effects of degree level education without the survey factor. This was an error and has been corrected in the revision p24, so the analysis for the effects of degree level education matches that conducted for gender (and presented directly above this section). Changes are highlighted, the main conclusions are unaltered.

Please can you provide clearer link to the data, analysis script and all materials used in the manuscript preferably one that is held in some repository like the OSF.

Response: The data are provided as the S2 Appendix to accompany the paper. We also routinely use the University of Nottingham Research Data Repository to make data freely available post-publication. These data have been deposited (linked to the title of the manuscript for now) with the reserved doi: 10.17639/nott.7305

---

## [Decision Letter · Decision Letter 1]

17 Jul 2023

PONE-D-23-08435R1Attitudes to the use of animals in biomedical research: Effects of stigma and selected research project summariesPLOS ONE

Dear Dr. Cassaday,

Thank you for submitting your manuscript to PLOS ONE. After careful consideration, we feel that it has merit but does not fully meet PLOS ONE’s publication criteria as it currently stands. Therefore, we invite you to submit a revised version of the manuscript that addresses the points raised during the review process.

Dear Professor Cassaday, your paper is much improved, as noted by the reviewers. I have re-read it and, after careful reflection, believe that Reviewer 1's comments could result in a measurable improvement to the paper. If you could address these changes to my satisfaction, I would be ready to give timely and positive consideration to your paper. I hope you will consider a re-submission.

We look forward to receiving your revised manuscript.

Kind regards,

Stefano Occhipinti

Academic Editor

PLOS ONE

Journal Requirements:

Reviewers' comments:

Reviewer's Responses to Questions

**Comments to the Author**

1. If the authors have adequately addressed your comments raised in a previous round of review and you feel that this manuscript is now acceptable for publication, you may indicate that here to bypass the “Comments to the Author” section, enter your conflict of interest statement in the “Confidential to Editor” section, and submit your "Accept" recommendation.

Reviewer #1: (No Response)

Reviewer #2: All comments have been addressed

2. Is the manuscript technically sound, and do the data support the conclusions?

Reviewer #1: Yes

Reviewer #2: Yes

3. Has the statistical analysis been performed appropriately and rigorously? 

Reviewer #1: Yes

Reviewer #2: Yes

4. Have the authors made all data underlying the findings in their manuscript fully available?

Reviewer #1: (No Response)

Reviewer #2: No

5. Is the manuscript presented in an intelligible fashion and written in standard English?

Reviewer #1: Yes

Reviewer #2: Yes

6. Review Comments to the Author

Reviewer #1: I thank the authors for the extensive changes they have made to the manuscript in response to comments from reviewers and the Editor. I have recommended some changes below which I believe will further strengthen the manuscript.

Introduction

• Some parts of the Introduction still need to be tightened up to more clearly tie the material presented to the study aims and hypotheses. For example, in the section discussing Stigma and blame (p5) there is a paragraph with general information about stigma, then stereotypes about people with mental illness, then biases towards specific disorders, followed by a short paragraph describing obesity stigma. As a reader I was left wondering why this information was being presented, and how it connected to the animal species section presented below it. I suggest combining paragraphs and adding in sentences tying the information presented to the aims and hypotheses to strengthen the argument being presented in the introduction.

Methods

• P10 suggest the change the phrasing of “(to avoid the possibility of responses to more than one survey)” to “reduce the possibility…” as it remains possible that people use more than one social network, so the risk can’t be avoided altogether.

• P10 it’s not clear to me what a matched social network would be

• On p11 in the design section the predictions are referred to as being both “key” and “general” – I don’t recall any key predictions being made in the introduction. Suggest using a consistent term.

Results

• P23, “largest p = 0.019 for the difference between” if the authors are aiming to comment on the largest effect, then it would be preferrable to compare effect sizes rather than p-values (this also applies elsewhere in the Results).

Discussion

• The first line of the discussion (p25) “The APQ ratings, which were…” would be strengthened by referring to the construct itself, rather than to APQ ratings (which tap into the construct of interest)

• P25 “and there was no dramatic difference in the profiles of responses” unclear to me what kind of difference would be considered “dramatic”; suggest re-wording

• In the manuscript “e.g.” is often used within a sentence (e.g., p32 “…wider range of disorders, e.g. depression and anxiety…”), the convention is typically e.g., within parentheses and “for example” (or similar) when part of a sentence.

• In the section discussing potential limitations and drop out rates for summaries the following sentence “This may be a real effect if participants were disengaged by the technical summary.” needs tightening up (i.e., it’s not clear what “real effect” means). Suggest also linking to the need for future research examining effects of disengaging due to technical summaries.

General

• There are some grammatical and typographic errors in the manuscript that will need correcting prior to publication

[Note that page numbers refer to marked changes doc]

Reviewer #2: Thanks for the detailed response to the reviewer comments. Currently the data is on the Nottingham depository but is not downloadable I trust this will be done upon acceptance of the manuscript although I was disappointed as reviewer this could not be accessed via a reviewer link. We have disagreement on the implications of multiple imputation which I would recommend the authors look into more deeply as it could have been good to use in this manuscript. Beyond those two set backs the manuscript is much improved with clear aims, hypothesis and analysis that is more sound.

7. PLOS authors have the option to publish the peer review history of their article (what does this mean?). If published, this will include your full peer review and any attached files.

Reviewer #1: No

Reviewer #2: No

---

## [Author Response · Author response to Decision Letter 1]

26 Jul 2023

Responses to reviewers

We thank the Editor and Reviewer for their constructive comments to further improve the manuscript. The points raised are addressed below and revisions to the manuscript are highlighted. Our own minor corrections are also highlighted.

Editor

Response: We have addressed Reviewer #1’s further comments as required (please see below). Reviewer #2 confirms that all comments have been addressed. I confirm that we submitted the data as a supplementary file which we assumed should have been available to reviewers. I put an embargo on the University of Nottingham repository link as we prefer to publish the findings before we share the underpinning data without any restriction. There is still a difference of opinion on the use of multiple imputation for missing data, but Reviewer 2 is overall satisfied.

Reviewer #1

 Introduction

• Some parts of the Introduction still need to be tightened up to more clearly tie the material presented to the study aims and hypotheses. For example, in the section discussing Stigma and blame (p5) there is a paragraph with general information about stigma, then stereotypes about people with mental illness, then biases towards specific disorders, followed by a short paragraph describing obesity stigma. As a reader I was left wondering why this information was being presented, and how it connected to the animal species section presented below it. I suggest combining paragraphs and adding in sentences tying the information presented to the aims and hypotheses to strengthen the argument being presented in the introduction.

Response: The Introduction has been tightened up as suggested.

Methods

• P10 suggest the change the phrasing of “(to avoid the possibility of responses to more than one survey)” to “reduce the possibility…” as it remains possible that people use more than one social network, so the risk can’t be avoided altogether.

Response: This has been done.

• P10 it’s not clear to me what a matched social network would be

Response: The social networks were judged to be similar based on the similar demographics of the researchers, but we agree that we cannot really claim that they were ‘matched’ in any formal sense. The wording has been revised accordingly.

• On p11 in the design section the predictions are referred to as being both “key” and “general” – I don’t recall any key predictions being made in the introduction. Suggest using a consistent term.

Response: We have removed this unhelpful distinction and now consistently refer to predictions.

Results

• P23, “largest p = 0.019 for the difference between” if the authors are aiming to comment on the largest effect, then it would be preferrable to compare effect sizes rather than p-values (this also applies elsewhere in the Results).

Response: The Results presentation has been edited to avoid this issue. We now simply refer to the S3 Appendix for full details of the pairwise comparisons. We have also added a couple of missing effect sizes for the ANOVAs.

Discussion

• The first line of the discussion (p25) “The APQ ratings, which were…” would be strengthened by referring to the construct itself, rather than to APQ ratings (which tap into the construct of interest)

Response: The opening to the Discussion has been re-phrased.

• P25 “and there was no dramatic difference in the profiles of responses” unclear to me what kind of difference would be considered “dramatic”; suggest re-wording

Response: This statement has been re-worded.

• In the manuscript “e.g.” is often used within a sentence (e.g., p32 “…wider range of disorders, e.g. depression and anxiety…”), the convention is typically e.g., within parentheses and “for example” (or similar) when part of a sentence.

Response: We have searched and replaced the use of ‘e.g.’ outside of parentheses as suggested.

• In the section discussing potential limitations and drop out rates for summaries the following sentence “This may be a real effect if participants were disengaged by the technical summary.” needs tightening up (i.e., it’s not clear what “real effect” means). Suggest also linking to the need for future research examining effects of disengaging due to technical summaries.

Response: This section has been clarified.

General

• There are some grammatical and typographic errors in the manuscript that will need correcting prior to publication

Response: We have run further checks for grammatical and typographical errors, which we have corrected and highlighted throughout.

---

## [Decision Letter · Decision Letter 2]

7 Aug 2023

Attitudes to the use of animals in biomedical research: Effects of stigma and selected research project summaries

PONE-D-23-08435R2

Dear Dr. Cassaday,

We’re pleased to inform you that your manuscript has been judged scientifically suitable for publication and will be formally accepted for publication once it meets all outstanding technical requirements.

Kind regards,

Ali B. Mahmoud, Ph.D.

Academic Editor

PLOS ONE

Additional Editor Comments (optional):

Reviewers' comments:

Reviewer's Responses to Questions

**Comments to the Author**

1. If the authors have adequately addressed your comments raised in a previous round of review and you feel that this manuscript is now acceptable for publication, you may indicate that here to bypass the “Comments to the Author” section, enter your conflict of interest statement in the “Confidential to Editor” section, and submit your "Accept" recommendation.

Reviewer #1: All comments have been addressed

Reviewer #2: All comments have been addressed

2. Is the manuscript technically sound, and do the data support the conclusions?

Reviewer #1: (No Response)

Reviewer #2: Yes

3. Has the statistical analysis been performed appropriately and rigorously? 

Reviewer #1: (No Response)

Reviewer #2: Yes

4. Have the authors made all data underlying the findings in their manuscript fully available?

Reviewer #1: (No Response)

Reviewer #2: Yes

5. Is the manuscript presented in an intelligible fashion and written in standard English?

Reviewer #1: (No Response)

Reviewer #2: Yes

6. Review Comments to the Author

Reviewer #1: (No Response)

Reviewer #2: It sounds and appears like the Authors have addressed reviewers one concerns. My only concern last time was around the availability of the data and it sounds like this has been addressed by uploading it to the repository. It is a shame that the repository does not have a reviewer only link as the OSF does possibly something for Nottingham to work on.

7. PLOS authors have the option to publish the peer review history of their article (what does this mean?). If published, this will include your full peer review and any attached files.

Reviewer #1: No

Reviewer #2: No

---

## [Editor Report · Acceptance letter]

10 Aug 2023

PONE-D-23-08435R2 

Attitudes to the use of animals in biomedical research: Effects of stigma and selected research project summaries 

Dear Dr. Cassaday:

I'm pleased to inform you that your manuscript has been deemed suitable for publication in PLOS ONE. Congratulations! Your manuscript is now with our production department. 

Kind regards, 

on behalf of

Dr. Ali B. Mahmoud 

Academic Editor

PLOS ONE